# Periplasmic oxidized-protein repair during copper stress in *E. coli*: A focus on the metallochaperone CusF

**Alexandra Vergnes**[1], **Camille Henry**[1¤], **Gaia Grassini**[1], **Laurent Loiseau**[1], **Sara El Hajj**[1], **Yann Denis**[2], **Anne Galinier**[1], **Didier Vertommen**[3], **Laurent Aussel**[1], **Benjamin Ezraty**[1]*

**1** Aix-Marseille University, CNRS, Laboratoire de Chimie Bactérienne, Institut de Microbiologie de la Méditerranée, Marseille, France, **2** Institut de Microbiologie de la Méditerranée, Plate-forme Transcriptomique, Marseille, France, **3** de Duve Institute, MASSPROT Platform, Université Catholique de Louvain, Brussels, Belgium

¤ Current address: Department of Biochemistry, University of Wisconsin-Madison, Madison, Wisconsin, United States of America

* ezraty@imm.cnrs.fr

**Data Availability Statement:** All relevant data are within the manuscript and its Supporting Information files.

## Abstract

Methionine residues are particularly sensitive to oxidation by reactive oxygen or chlorine species (ROS/RCS), leading to the appearance of methionine sulfoxide in proteins. This post-translational oxidation can be reversed by omnipresent protein repair pathways involving methionine sulfoxide reductases (Msr). In the periplasm of *Escherichia coli*, the enzymatic system MsrPQ, whose expression is triggered by the RCS, controls the redox status of methionine residues. Here we report that MsrPQ synthesis is also induced by copper stress via the CusSR two-component system, and that MsrPQ plays a role in copper homeostasis by maintaining the activity of the copper efflux pump, CusCFBA. Genetic and biochemical evidence suggest the metallochaperone CusF is the substrate of MsrPQ and our study reveals that CusF methionines are redox sensitive and can be restored by MsrPQ. Thus, the evolution of a CusSR-dependent synthesis of MsrPQ allows conservation of copper homeostasis under aerobic conditions by maintenance of the reduced state of Met residues in copper-trafficking proteins.

## Author summary

This study investigates the interconnection between the copper stress response and the methionine redox homeostasis in the Gram-negative bacterium *Escherichia coli*. We report that the copper-activation of the CusSR two-component system induces the expression of the genes encoding the periplasmic oxidized-protein repair system, MsrPQ. This repair system was shown to be crucial for CusCFBA copper efflux pump activity under aerobic conditions as it maintains the periplasmic component CusF in its functional reduced form. Methionine emerges as a critical residue in copper trafficking proteins. However, its high affinity for metals is counterbalanced by its high susceptibility to

**Funding:** BE was supported by Agence Nationale Recherche (ANR) (#ANR-16-CE11-0012-02 METOXIC), (#ANR-21-CE44-0024 MetCop) and Centre National Recherche Scientifique (CNRS) (#PICS-PROTOX). CH was supported by Fondation pour la Recherche Médicale (FRM). GG was supported by Aix-Marseille Université (AMidex-Post-doc). The funders had no role in study design, data collection and analysis, decision to publish, or preparation of the manuscript.

**Competing interests:** The authors have declared that no competing interests exist.

oxidation. Therefore, the induction of *msrPQ* by copper allows copper homeostasis under aerobic conditions, illustrating that *E. coli* has developed an integrated and dynamic circuit for sensing and counteracting stress caused by copper and oxidants, thus allowing bacteria to adapt to host cellular defences.

## Introduction

Accumulation of damaged proteins hampers biological processes and can lead to cellular dysfunction and death. Chaperones, proteases and repair enzymes allow cells to confront these challenges and regulate protein homeostasis. The activity of these protein families defines "protein quality control" [1] and under stress conditions (high temperatures, oxidative or metal stress), signal transduction cascades up-regulate protein quality control to reduce the appearance of aggregation-prone molecules [2]. Protein quality control is also involved in housekeeping functions in different cellular compartments throughout the cellular life cycle. Within proteins, sulfur-containing amino acids such as methionine (Met) are targets for reactive oxygen species (ROS) and reactive chlorine species (RCS), the latter being more efficient at converting Met to its oxidized form, methionine sulfoxide (Met-O) [3]. This oxidation reaction is reversible due to the action of methionine sulfoxide reductases (Msr) [4]. MsrPQ in *E. coli* is an Msr system necessary for periplasmic proteins quality control, in which MsrP reduces Met-O and MsrQ is the membrane-bound partner required for MsrP activity [5]. We have previously shown that the *msrPQ* genes are found in an operon with the *hiuH* gene (encoding for a 5-hydroxyisourate (5-HIU) hydrolase [6]), and that *hiuH-msrPQ* expression was induced by RCS (HOCl) in an HprSR-dependent manner [5,7]. HprSR is a two-component system (TCS), in which HprS is a histidine kinase (HK) sensor and HpsR the cytoplasmic response regulator (RR) [7]. The periplasmic chaperone SurA is one of the preferred substrates of MsrP [5] and proteomic studies have pinpointed processes including metal homeostasis (FecB, RcnB and ZnuA), under the supervision of MsrP [5].

Copper is an essential prosthetic group in major *E. coli* enzymes, including cytochrome *bo* quinol oxidase and copper-zinc superoxide dismutase, however, high copper concentrations are toxic to the cell [8]. In aerobiosis, copper toxicity may be due to its involvement in the Fenton-like reactions which generate the highly reactive hydroxyl radicals (HO˙) [9]. The Imlay group showed that the copper-mediated Fenton reaction does not cause oxidative DNA damage in *E. coli* cytoplasm [10]. Conversely, copper EPR spectroscopy suggested that most of the copper-mediated HO˙ formation does not occur near DNA, but in the periplasmic compartment [10]. Copper is more toxic under anaerobic conditions due to the predominant presence of Cu(I) [11] and Fe-S clusters are the main intracellular targets of copper toxicity, even in the absence of oxygen [12]. Moreover, a recent study shows that copper induces protein aggregation, which represents a central mechanism of Cu toxicity under anaerobic conditions [13].

Regulation of copper homeostasis is therefore required to maintain intracellular copper at low levels [14]. In *E. coli*, at least three systems are involved in copper tolerance: (i) CopA, a P-type ATPase which pumps copper from the cytoplasm to the periplasm [15]; (ii) CueO, a periplasmic multi-copper oxidase that oxidizes Cu(I) to the less toxic Cu(II) [16,17] and (iii) CusCFBA, an RND-type (resistance, nodulation, division) efflux pump responsible for extrusion of copper into the extracellular environment [18]. This RND-type efflux pump consists of CusA, the inner-membrane proton antiporter, CusB, the periplasmic protein, CusC the outer-membrane protein, and CusF, the periplasmic metallochaperone that supplies copper to the pump. The *cusCFBA* operon is under the control of the CusSR pathway in which CusS is the

sensor and CusR the RR [19]. Finally the CueR transcriptional regulator regulates both *copA* and *cueO* expression [20].

Several lines of evidence point to the role of methionine residues in copper coordination within proteins such as CopA, CueO, CusF and CusAB proteins [8,21]. Mutation of the conserved Met204 in CopA yields an enzyme with a lower turnover rate, which is explained by a decrease in Cu(I) transfer efficiency from CopA to the chaperone CusF [22]. CueO has a methionine-rich helix which allows Cu(I) binding to provide a cuprous oxidase function [16,17,23]. The periplasmic copper chaperone CusF binds Cu(I) via two important methionine residues [18,24,25]. Also for the periplasmic adaptor CusB, and the inner membrane component CusA, methionine residues play a pivotal part in Cu(I) binding and in the stepwise shuttle mechanism by which the pump extrudes copper from the cell [26–29]. In summary, in many cases Met residues have been identified as crucial for copper resistance.

ROS/RCS could impair the detoxification function of CueO, CopA and CusCFBA through the oxidation of Met residues; MsrP would then be required to reduce Met-O to allow proteins to recover their copper homeostatic functions. This postulate is reinforced by a study showing that the CusSR system up-regulates the expression of the *hiuH* gene, located upstream of *msrP* [30,31], opening up the possibility that MsrP is produced during copper stress to maintain at least one of the three systems involved in copper tolerance. Here, we report that *msrP* is induced during copper stress via CusSR. We have also established, by a phenotypic approach, that MsrP was crucial for maintaining CusCFBA pump activity under aerobic conditions. By focusing on the periplasmic proteins CusB and CusF, we have demonstrated that the metallo-chaperone underwent post-translational Met-O modification after $H_2O_2$ treatment, affecting its activity, which can be restored by MsrPQ.

## Results

### *msrP* expression is induced by $CuSO_4$ in a CusSR-dependent manner

We have recently shown that the *E. coli* genes *hiuH*, *msrP* and *msrQ* belong to the same operon [7] and previous studies have indicated that copper induces *hiuH* expression (Fig 1A) [30]. To corroborate these observations, the role of copper in the production of MsrP was investigated. To measure the effect of copper on the *hiuH-msrPQ* operon, quantitative reverse transcription polymerase chain reaction (qRT-PCR) experiments were performed in a wild-type strain of *E. coli* cultured in LB under aerobic conditions. *hiuH*, *msrP* and *msrQ* mRNA levels were found to increase significantly in copper-treated cells ($\sim 190$, $\sim 13$ and $\sim 30$-fold respectively) (Fig 1B). Under the same growth condition, western blot analyses showed higher MsrP protein levels following $CuSO_4$ treatment (Fig 1C). Next, we wondered whether the TCS HprSR or CusSR regulated the expression of the *hiuH-msrPQ* operon under copper stress. The translational *msrP-lacZ* reporter fusion was then introduced into the wild-type, $\Delta hprRS$ and $\Delta cusRS$ strains for ß-galactosidase assays. The strains carrying the chromosomal reporter fusion were cultured in M9/CASA medium in the absence or presence of 500 μM of $CuSO_4$. The *msrP-lacZ* activity increased ($\approx$ 4-fold) after exposure to $CuSO_4$ in the wild-type and $\Delta hprRS$ strains, but not in the $\Delta cusRS$ strain. The increase in *msrP* expression following copper exposure was shown to be dependent on CusSR but not on HprSR (Fig 1D), consistently with previous reports [30]. The copper-dependent induction of *msrP* expression was lower than HOCl-HprSR dependent induction, in which the cells exhibited $\approx$ 60-fold higher ß-galactosidase activity (Fig 1E and [5,7]). Interestingly, the HOCl/copper combination gave the same level of *msrP* expression as HOCl alone (Fig 1E). These results show that MsrP concentrations increase in response to copper in a CusSR-dependent manner.

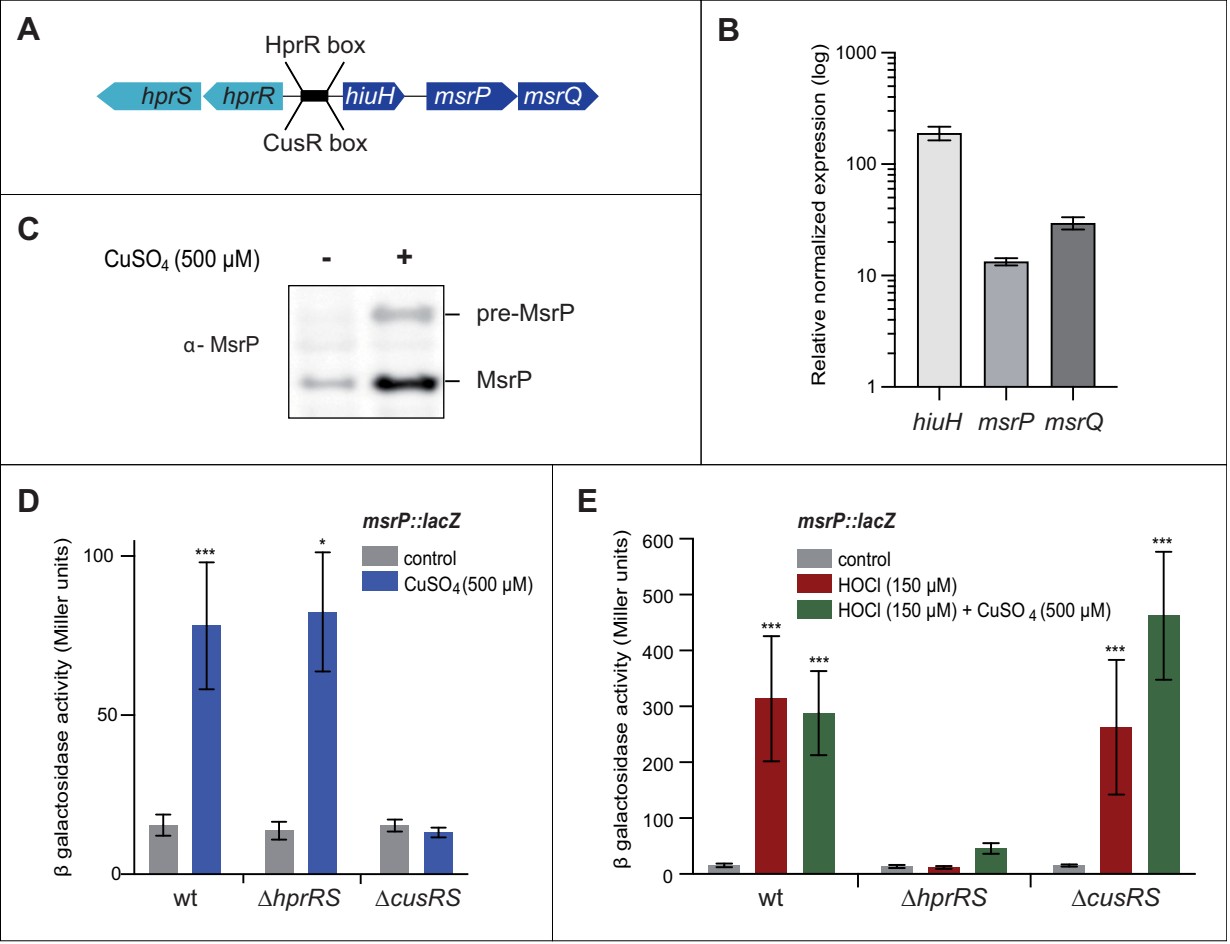

**Fig 1. Copper regulates *msrP* expression in a CusSR-dependent manner. A)** Schematic representation of the *hiuH-msrPQ* operon in *E. coli*. The intergenic region between *hiuH* and *hprR* contains the CusR and HprR boxes (CATTACAAAATTGTAATG) [30]. **B)** Relative normalized expression of *hiuH*, *msrP* and *msrQ* genes during copper stress. RNA was extracted from the wild-type strain grown in LB with CuSO$_4$ (500 μM) to an OD$_{600nm}$ ≈ 2. Quantitative real-time PCR was performed to amplify the *hiuH*, *msrP* and *msrQ* genes. Results are the means ± standard deviation of three independent experiments. **C)** Immunoblot analysis using an anti-MsrP antibody, showing the production of MsrP by CuSO$_4$ (500 μM) stress from the wild-type strain grown in LB. The image is representative of experiments carried out in triplicate. **D-E)** *msrP::lacZ* fusion was used as a proxy for *msrP* expression. Wild-type, Δ*hprRS* and Δ*cusRS* strains were grown in M9/CASA medium with or without the addition of 500 μM CuSO$_4$ (**D**) or 150 μM HOCl or a combination of CuSO$_4$ (500 μM) and HOCl (150 μM) (**E**), and ß-galactosidase assays were performed. Deletion of *cusRS* prevents *msrP* induction by copper, whereas deletion of *hprRS* prevents its induction by HOCl. Error bars, mean +/- s.e.m.; n = 8 for wild-type and Δ*cusRS*, n = 3 for Δ*hprRS*. Asterisks indicate a statistically significant difference between control and stressed conditions. *P ≤ 0.05; **P ≤ 0.01; and ***P ≤ 0.001 (Mann-Whitney U test).

## MsrP is required for copper tolerance

We hypothesized that MsrP might be important for cell growth when copper availability is high as *hiuH-msrPQ* is part of the CusSR regulon ([30] and previous paragraph). To test this, the Δ*msrP* strain was exposed to copper stress. The growth of the *msrP* mutant strain was first assayed on M9 plates containing CuSO$_4$ (12.5 to 20 μM). Disruption of *msrP* did not lead to significant copper sensitivity compared to a wild type strain under aerobic growth conditions (Fig 2A). We reasoned that functional redundancy between copper homeostasis systems might mask the importance of MsrP in copper tolerance (Fig 2B) and therefore, focused on MsrP and the CusCFBA efflux pump, as they are both part of the CusSR-mediated response. To test this hypothesis, the Δ*copA* and Δ*cueO* mutations were introduced into the Δ*msrP* mutant. The

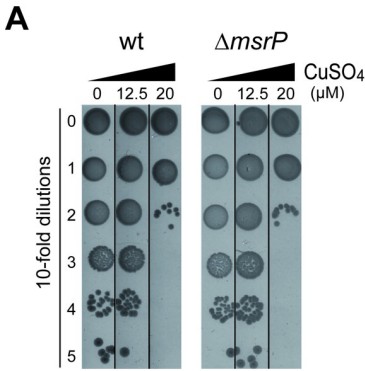

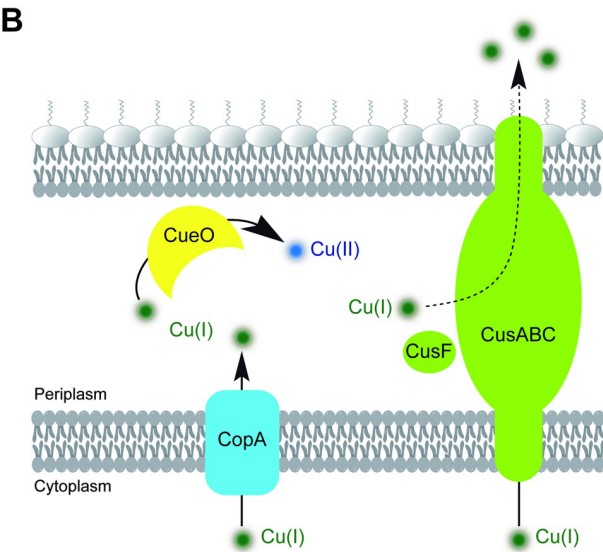

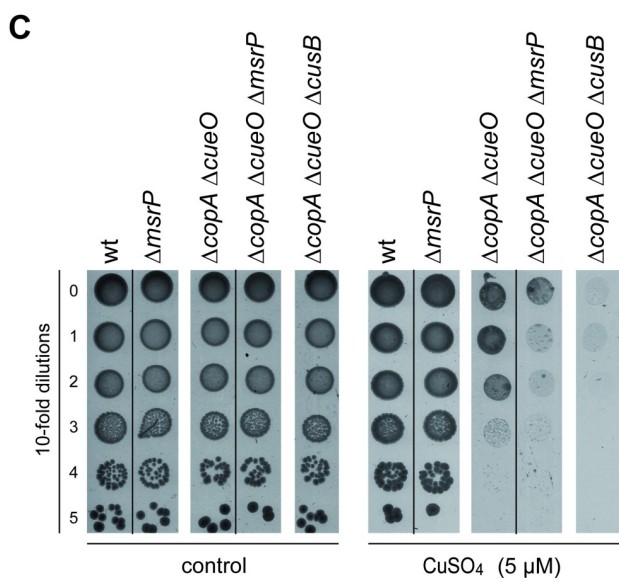

**Fig 2. Involvement of MsrP in copper tolerance in *E. coli*. A)** Plating efficiency of wild-type and Δ*msrP* strains in the presence of CuSO$_4$. Cells were grown to an exponential phase (OD$_{600nm}$ ≈ 0.1) at 37˚C in M9 medium and 10-fold serial dilutions were spotted onto M9 plates, with or without the addition of CuSO$_4$ at the concentrations given (top panel). No significant difference was observed between either strains. **B)** Schematic view of the copper homeostasis systems in *E. coli*. CopA (in blue) translocates Cu(I) ions from the cytoplasm into the periplasm. CueO (in yellow) oxidizes Cu(I) ions to the less toxic form Cu(II). CusCBA efflux system (in green) pumps out copper to the extracellular environment. The CusF protein (in green), part of the *cusCFBA* operon, is a periplasmic metallochaperone which supplies copper to the pump. **C)** Plating efficiency of wild-type, Δ*msrP*, *(ΔcopA ΔcueO)*, *(ΔcopA ΔcueO ΔmsrP)* and *(ΔcopA ΔcueO ΔcusB)* strains in the presence of CuSO$_4$. Cells were grown to an exponential growth phase (OD$_{600nm}$ ≈ 0.1) at 37˚C in M9 medium and 10-fold serial dilutions were spotted onto M9 plates, without stress (left side panel) and with CuSO$_4$ (5 μM)(right side panel). The images are representative of experiments carried out at least three times.

copper sensitivity of this triple mutant was monitored on M9 plates containing 5 μM of CuSO$_4$. The Δ*copA* Δ*cueO* Δ*msrP* strain was shown to be more sensitive to copper than the parental Δ*copA* Δ*cueO*, MsrP proficient strain (Fig 2C). The copper sensitivity of the Δ*copA* Δ*cueO* Δ*msrP* strain was found to be similar to that of the triple copper tolerance system mutant: Δ*copA* Δ*cueO* Δ*cusB* strain. These results suggest that MsrP may play a role in copper tolerance.

## The copper sensitivity of Δ*copA* Δ*cueO* Δ*msrP* is oxygen-dependent

The above findings suggested that the periplasmic oxidized-protein repair system is part of the copper stress response. Thus, we investigated the possibility that the link between MsrP and copper was oxidative stress dependent by performing the copper sensitivity assay under anaerobic conditions. While we observed a copper sensitivity of the Δ*copA* Δ*cueO* mutant in the presence of 5 μM CuSO$_4$ under aerobic conditions, 1.5 μM CuSO$_4$ was sufficient to cause the same effects under anaerobic conditions due to the higher toxicity of copper in the absence of oxygen. In doing so, we did not detect copper-dependent growth inhibition of the Δ*copA* Δ*cueO* Δ*msrP* mutant compared to the isogenic parental MsrP- proficient strain (Fig 3A). To

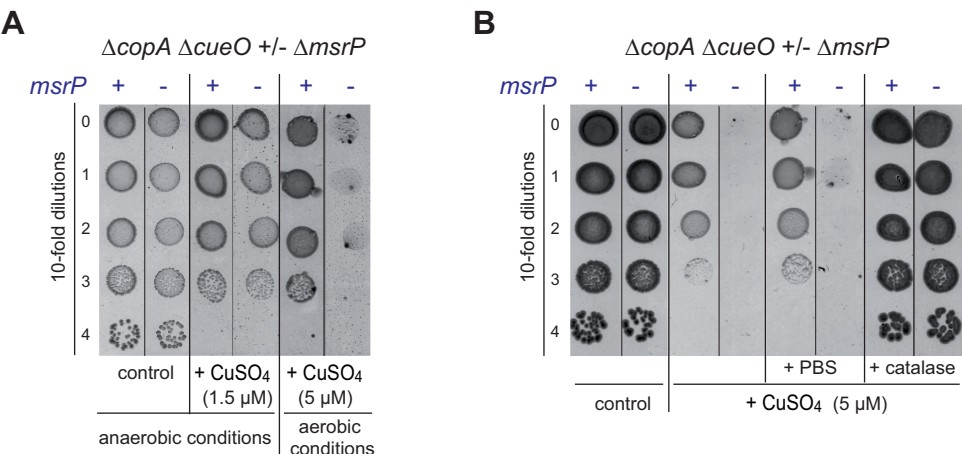

**Fig 3. The copper hypersensitivity of the Δ*copA* Δ*cueO* Δ*msrP* strain is ROS dependent. A)** Plating efficiency of Δ*copA* Δ*cueO* and Δ*copA* Δ*cueO* Δ*msrP* strains onto M9 plates in the presence of CuSO$_4$ (1.5 μM) under anaerobic conditions and in the presence of CuSO$_4$ (5 μM) under aerobic conditions. The same protocol as described in Fig 2 was used, except that plates were incubated in the absence of oxygen for 4 days. **B)** Plating efficiency of Δ*copA* Δ*cueO* and Δ*copA* Δ*cueO* Δ*msrP strains* onto M9 plates in the presence of CuSO$_4$ (5 μM) and catalase (2,000 units). 10-fold serial dilutions were spotted onto M9 plates with or without CuSO$_4$ in the presence of phosphate buffered saline (PBS) as a control or catalase under aerobic conditions. The images are representative of experiments carried out at least three times.

obtain more direct evidence that ROS are involved in the copper sensitivity of the strain lacking MsrP, an excess of catalase was added to plates before cell spreading—this method has been shown to reduce $H_2O_2$ levels under aerobic conditions [32]. Adding catalase to plates eliminated the *msrP*-deficient strain phenotype (Fig 3B). These data are consistent with the copper sensitivity of the Δ*copA* Δ*cueO* Δ*msrP* strain being ROS dependent.

## MsrP is required for copper tolerance by maintaining CusF activity

The above findings suggested that one or more components of the copper-efflux system CusCFBA may be damaged by oxidation. MsrP could therefore be essential for maintaining the CusCFBA pump in a reduced state. One prediction of our model is that CusCFBA pump overproduction should compensate for reduced efflux due to oxidation. To test this, the pCusCFBA plasmid encoding the whole operon was used, showing that the copper hypersensitivity of the Δ*copA* Δ*cueO* Δ*msrP* strain could be suppressed upon overexpression of the *cusCFBA* operon (Fig 4). In addition, the overexpression of *cusCFBA* genes was found to be slightly harmful to the cell, even in the absence of copper (Fig 4). To further test the prediction and to identify the limiting periplasmic component of the pump, the two periplasmic subunits CusB and CusF were expressed separately. The overproduction of CusB, but not CusF, was observed to be toxic to the cell (Fig 4). Interestingly, CusF overexpression in the Δ*copA* Δ*cueO* Δ*msrP* strain suppressed copper sensitivity of this strain (Fig 4). In spite of our efforts to find a more discriminate assay, the difference between the mutant and the MsrP proficient strain were best observed on the agar-containing copper plate assay. These results provide evidence that MsrP is at least involved in maintaining CusF activity.

## *In vivo* evidence for the consequences of CusF oxidation, using CusF^{M47Q/M49Q} as a proxy for of Met47 and Met49 oxidation

CusF is a soluble periplasmic protein that transfers copper directly to the CusCBA pump. The mature-CusF form contains four methionine residues (Met8, Met47, Met49 and Met59) of which Met47 and Met49, in addition to His36, are used as copper coordination ligands, with a nearby tryptophan (Trp44) capping the metal site [24]. Analysis of the apo-CusF structure showed that Met47 and Met49 are exposed to the solvent, with the sulfur atoms accessible,

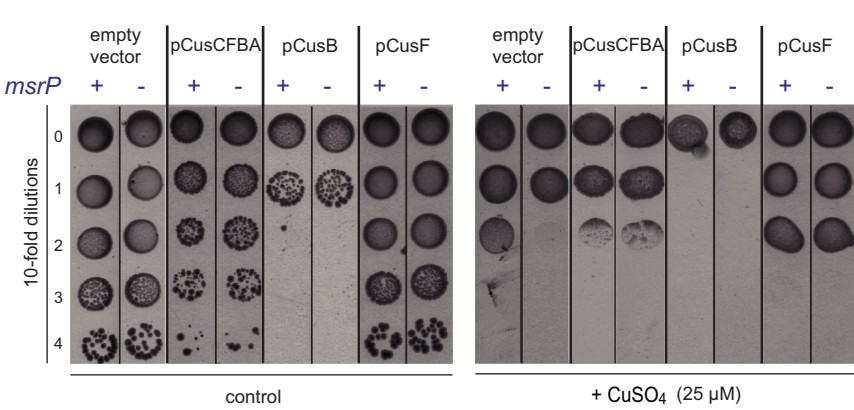

**Fig 4. Overexpression of CusF suppressed the copper hypersensitivity of the Δ*copA* Δ*cueO* Δ*msrP* strain.** Plating efficiency of the Δ*copA* Δ*cueO* Δ*msrP* strain carrying empty vector, pCusCFBA, pCusB or pCusF onto M9 plates in the presence of CuSO₄ (25 μM). The same protocol as described in Fig 2 was used, except plates contained ampicillin (50 μg/ml) and IPTG (100 μM). The images are representative of experiments carried out at least three times.

whereas in its copper-bound form, CusF underwent a conformational change whereby the sulfur of Met49 became inaccessible while Met47 appears to remain on the surface (Fig 5A) [24,33]. We hypothesized that Met47 and Met49 oxidation could impair CusF activity and replaced these Met residues by Ile (I) or Gln (Q), the latter mimicking Met oxidation [34]. The copper sensitivity of the ΔcopA ΔcueO ΔcusF strain was exploited to assess the activity of the CusF variants by trans-complementation with the mutated genes (Fig 5B). The phenotype of the ΔcopA ΔcueO ΔcusF strain was complemented in trans by the gene expressing wild-type CusF. Conversely, expression of the CusF[M47I/M49I] did not complement the strain as previously reported [18]. The CusF[M47Q/M49Q] variant partially complemented the copper sensitivity of the ΔcusF strain but not as well as wild-type CusF (Fig 5B) suggesting that Met47 and Met49 oxidation could hamper CusF activity.

## Methionine oxidation of CusF gives rise to non-functional protein

We sought to characterize the metal-binding capacity of the CusF oxidized form. First, purified CusF protein was treated with $H_2O_2$ (50 mM) for 2 hours and analysed by mass spectrometry. CusF oxidation reaction was first monitored by gel-shift assays (by SDS-polyacrylamide gel electrophoresis), as Met-O-containing proteins ran slower than their reduced counterparts, leading to a mobility shift (Fig 5C- upper panel) [35]. Met residues present in the mature CusF were identified in peptides detectable by mass spectrometry after trypsin digestion. Met47 and Met49 are part of the same peptide and therefore, it was not possible to determine the oxidation level of each residue separately. The Met47-Met49 containing peptide from untreated protein had around 25% of Met present as the Met-O form. This basal level of protein oxidation is commonly obtained and is usually assigned to the trypsin digestion protocol [36]. After $H_2O_2$ treatment, the proportion of Met-O increased to 98.9% for Met47- and Met49-containing peptides (Fig 5C-lower panel).

CusF has been shown to bind Cu(I) and Ag(I) with similar protein coordination chemistry [37,38]. We took advantage of the Ag(I)-binding property of CusF to assess the metal-binding capacities of the oxidized forms of CusF using AgNO$_3$, instead of the highly toxic Cu(I) generation systems. For this, the intrinsic fluorescence of CusF was monitored: Trp fluorescence emission peaks at 350 nm and the addition of increasing amounts of AgNO$_3$ to CusF led to progressive fluorescence quenching (with a maximum of 77%).

Compared to the native form of CusF, fluorescence quenching was strongly reduced with the oxidized form of CusF (CusF[ox]): only a slight decrease in intrinsic fluorescence was observed, even at the highest AgNO$_3$ concentration tested (maximal fluorescence quenching = 11%). The difference could be due to the alteration of the Met residues involved in metal coordination ligands. To test this hypothesis, the non-functional CusF[M47Q/M49Q] variant was purified (substitutions mimicking Met47 and Met 49 oxidation), and its intrinsic fluorescence was measured. Fluorescence quenching for CusF[M47Q/M49Q] was comparable to that of CusF[ox] (maximal fluorescence quenching of 10%), indicating that CusF oxidation affected the metal-binding capacity of the protein (Fig 5D). The low fluorescence quenching of CusF[ox] and the mutated variant were supposed to be due to a Met-independent metal-binding domain. The fact that Trp residue emission was less affected by AgNO$_3$ for both the oxidized and the mutated M to Q forms of CusF in comparison to the native protein could result from a local conformational change, leading to a non-functional protein. However, structural analysis of CusF, CusF[ox] and CusF[M47Q/M49Q] by circular dichroism (CD) spectroscopy revealed no difference between the CD spectra obtained with these three forms, i.e. neither $H_2O_2$ oxidation nor M to Q mutations affected the overall conformation of the protein (S1 Fig). In order to test the reversibility of CusF oxidation, the CusF[ox] was treated with MsrP enzyme in the presence of a

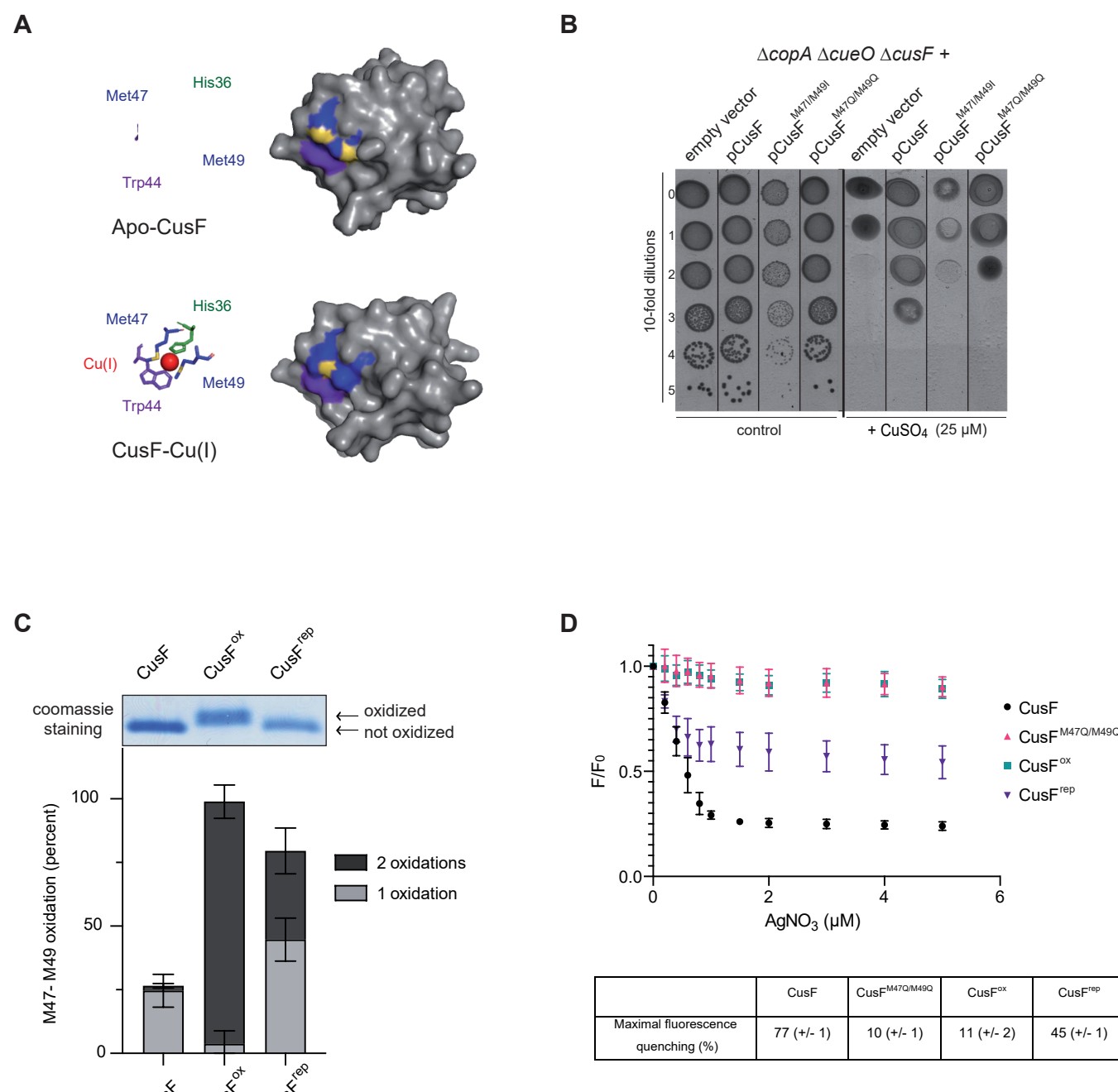

**Fig 5. Methionine oxidation of CusF is deleterious. A)** Aligned structures of the *E. coli* apo-CusF and CusF-Cu(I) adapted from PDB:1ZEQ and 2VB2 respectively [24,33] with stick and surface representations of CusF. Residues His 36 (green), Met 47, Met 49 (blue with sulphur atoms highlighted in yellow) and Trp44 (purple) are shown. The Cu(I) ion is shown in red. **B)** Plating efficiency of the $\Delta copA$ $\Delta cueO$ $\Delta cusF$ strain carrying empty vector, pCusF, pCusF$^{M47I/M49I}$ or pCusF$^{M47Q/M49Q}$ vectors onto M9 plates in the presence of CuSO$_4$ (25 μM). The same protocol as described for Fig 2 was used, except plates contained ampicillin (50 μg/ml) and IPTG (50 μM). The images are representative of experiments carried out at least three times. **C)** Gel shift assay and mass spectrometry relative quantification by LFQ of the oxidation of Met47 and Met49. **D)** Silver binding analysed by quenching of intrinsic tryptophan fluorescence. Increasing concentrations of AgNO$_3$ (0, 0.2, 0.4, 0.6, 0.8, 1, 1.5, 2, 3, 4, and 5 μM) were added to 1 μM CusF, CusF$^{M47Q/M49Q}$, CusF$^{ox}$ and CusF$^{rep}$. The emission spectrum of CusF was recorded after each addition as described in the Materials and Methods. The integrated fluorescence peak (between 300 and 384 nm) in the presence of AgNO$_3$ (F) was compared with the peak obtained in its absence (F$_0$). The F/ F$_0$ ratio was plotted against the concentration of AgNO$_3$, after correction for the inner filter effect of AgNO$_3$ measured on *N*-acetyltryptophanamide (NATA). The maximal fluorescence quenching for each variant of CusF was reported as a percentage in the table.

reducing system (dithionite and benzyl viologen) to yield the repaired form (CusF$^{rep}$). Mass spectrometry and gel-shift assay of CusF$^{rep}$ revealed a decrease in Met-O content, showing partial repair of CusF$^{ox}$ (Fig 5C). Fluorescence quenching was also partially restored (maximal fluorescence quenching of 45%, versus 77% for the native protein), probably reflecting a mix of oxidized and repaired CusF forms, in which the Trp residue had returned to its initial conformation (Fig 5D). In conclusion, oxidized CusF is non-functional and MsrP can restore CusF activity by reducing Met-O.

## Discussion

Methionine has emerged as a critical residue in copper trafficking proteins, providing binding sites that allow metal transfer. However its high affinity for metals is counterbalanced by its high susceptibility to oxidation. Indeed, under oxidative conditions (ROS, RCS), methionine is one of the preferred oxidation targets in proteins [39]. However, methionine oxidation is reversible due to the universal presence of the methionine sulfoxide reductases (MSR), which reduce oxidized methionine residues [3]. Here, we have demonstrated that in *E. coli*, the presence of copper induces the expression of the *msrP* gene encoding the enzyme involved in the repair of periplasmic oxidized proteins. Phenotypic analysis under aerobic conditions demonstrated the role of MsrP in maintaining the CusCFBA copper export pump. Genetic and biochemical analyses provided evidence that the oxidation of the CusF copper chaperone, at the very least, leads to the loss of function of this pump. In summary, (i) deletion of *msrP* is detrimental to CusCFBA activity, (ii) overexpression of *cusF* suppresses this phenotype, (iii) oxidized CusF contains Met-O residues, (iv) oxidized CusF is inactive as is mutated CusF M47Q/M49Q and (v) MsrP reduces Met-O in CusF and restores its activity.

Interestingly, the *msrPQ* and *cusCFBA* operons were both shown to be regulated by the CusSR TCS during copper stress. The existence of a common regulatory pathway for *msrP* and *cusF* reinforces the idea of a functional link between both proteins (Fig 6). However, the possibility that other components of the Cus pump are targeted by ROS/RCS cannot be excluded, as well as the CopA and CueO proteins, which also contain methionine-rich sites involved in copper binding. Testing this hypothesis will be a field of future research.

In this study, *msrP* was shown to be expressed in the presence of copper in *E. coli*. This observation could be explained by the fact that copper participates in methionine oxidation via the copper-based Fenton reaction in the periplasm, like the analogous reaction driven by iron in the cytoplasm [10,40]. Our results reinforce this notion by demonstrating that even a protein involved in copper tolerance such as CusF is an oxidation target.

The *hiuH* gene, part of the *msrPQ* operon [7], encodes for a 5-hydroxyisourate (5-HIU) hydrolase, a protein involved in the purine catabolic pathway [41]. This enzyme catalyses the conversion of 5-HIU, a degradation product of uric acid, into 2-oxo-4-hydroxy-4-carboxy-5-ureidoimidazoline (OHCU). Based on the fact that copper has been shown to strongly inhibit the HiuH activity of *Salmonella* [6], Urano *et al.* proposed that the copper–dependent transcriptional regulation of *hiuH* might be important in maintaining uric acid metabolism [31]. Uric acid is generally considered to be an antioxidant having a free radical scavenging activity, but an opposite role as a copper-dependent pro-oxidant has also been reported [42]. Consequently, another hypothesis is that the copper and ROS/RCS up-regulation of *hiuH* might have a physiological role during oxidative stress. HprR and CusR have been shown to have the same recognition sequence and can bind to the consensus box with different affinities [31], leading to a collaborative or competitive interplay depending on the concentration of regulatory proteins. Therefore, a better characterization of the cross-regulation (copper *versus* oxidative stress) by the two TCS appears necessary. ROS/RCS and copper stresses are

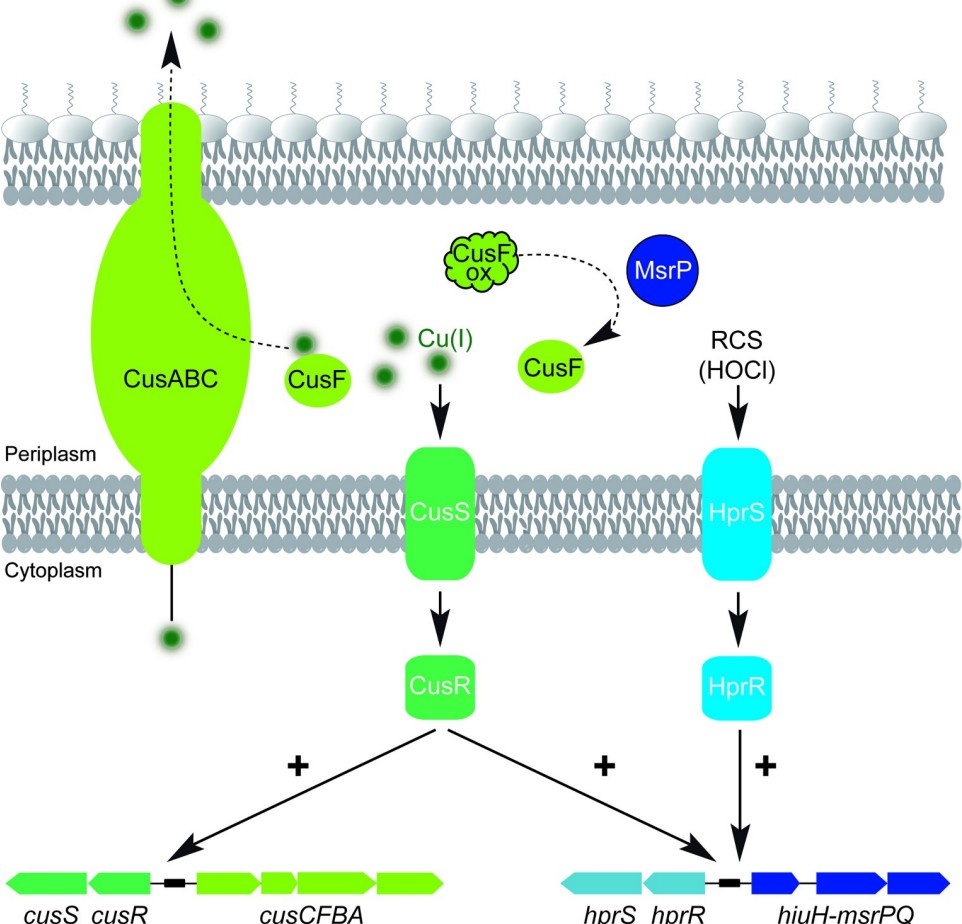

**Fig 6. Copper efflux pump and oxidized-protein repair system are co-regulated.** A working model illustrating the co-regulation of the copper efflux pump CusCFBA and the oxidized-protein repair system MsrPQ. Upon exposure to reactive chlorine species (RCS), the HprSR two-component system is activated leading to the up-regulation of the *hiuH-msrPQ* operon. Whereas, upon exposure to copper, the CusSR two-component system is activated leading to the up-regulation of the *cusCFAB* and *hiuH-msrPQ* operons. MsrP plays a role in copper homeostasis by controlling the redox status of methionine residues in the periplasmic metallochaperone CusF. CusF supplies copper to the efflux pump CusCBA, which then extrudes copper to the extracellular environment. By maintaining Met residues in a reduced form, MsrP appears to be essential for copper tolerance.

encountered during host-pathogen interactions [43]. During infection, phagocytic cells such as neutrophils produce ROS/RCS through NADPH oxidase and myeloperoxidase and accumulate copper in their phagosome via the ATP7A pump [44]. Thus, pathogens face both stresses at the same time. The interconnection between antimicrobial compounds produced by the immune system, like copper and HOCl, are an under-explored subject. Recently, the Gray laboratory reported that copper protects *E. coli* against killing by HOCl [45]. They identified the Cu(II) reductase RclA, which is induced by HOCl stress, as a central HOCl/copper combination resistance actor. The authors proposed that RclA prevents the formation of highly reactive Cu(III) by limiting the amount of Cu(II). Therefore, copper redox chemistry appears to be critical in the interaction between bacteria and the innate immune system. Interestingly, our study shows that *E. coli* has developed an integrated and dynamic circuit to sense and resist the combinatorial stresses caused by copper and HOCl, thus conferring an

important adaptive capacity to host cellular defences. Our findings will be confirmed by future investigations examining the interplay between copper/HOCl stresses and protein oxidation during pathogenesis.

## Materials and methods

### Strains and microbial techniques

The strains used in this study are listed in Table 1. The corresponding alleles of the deletion mutants were transferred from the Keio collection strain into the MG1655 wild-type strain by P1 transduction standard procedure and checked by PCR. The *hprRS* deletion mutant (strain CH100) was generated using a PCR knockout method developed by Datsenko and Wanner [46]. Briefly, a DNA fragment containing the *cat* gene flanked by the homologous sequences found upstream of the *hprR* gene and downstream of the *hprS* gene was PCR-amplified using pKD3 as template and the oligonucleotides *P1_Up_YedW(HprR)* and *P2_Down_YedV(HprS)*. The fragment was transformed into strain MG1655, carrying plasmid pKD46, by electroporation. Chloramphenicol-resistant clones were selected and verified by PCR. The same procedure was used for the *cusRS* deletion mutant (strain GG100) with the oligonucleotides *cusS_kan_rev* and *cusR_kan_for*. Primer sequences used in this study are listed in Table 2.

### Plasmid construction

The plasmids used in this study are listed in Table 3. The CusCFBA (IPTG induced) expression vector was constructed by amplifying the *cusCFBA* operon was amplified from the chromosome (MG1655) using primers *cusC-EcoRI fwd* and *cusA-XhoI-rev*. The resulting PCR product was cloned into PJF119EH using *EcoRI* and *XhoI/SalI* restriction sites, generating plasmid pAV79.

The CusF (IPTG induced) expression vector was constructed by amplifying the *cusF* gene from the chromosome (MG1655) using primers *cusF-EcoRI fwd* and *cusF-strep-HindIII rev*, which resulted in the fusion of a Strep-tag II coding sequence at the 3' end. The PCR product was cloned into PJF119EH using *EcoRI* and *HindIII* restriction sites, generating plasmid pAV54. The CusB (IPTG-induced) expression vector was constructed using the same procedure, using primers *cusB-EcoRI fwd and cusB-strep-HindIII rev* and generating plasmid pAV67.

### *cusF* directed mutagenesis

50 μl PCR reactions were performed using Q5 Hot start High-Fidelity DNA polymerase (New England Biolabs), PJF119EH-cusF (pAV54) as the template and primers *cusF fwd* and

**Table 1. Strains used in this study.** This table contains the information regarding the strains used in this study, including strain names, genotypes, description and source.

| Strain | Genotype and description | Source |
|---|---|---|
| MG1655 | WT | Laboratory collection |
| BE107 | MG1655 Δ*msrP*::Kan$^r$ | Gennaris *et al.* [5] |
| CH183 | MG1655 *msrP*::*lacZ* | Gennaris *et al.* [5] |
| CH100 | MG1655 *msrP*::*lacZ* Δ*hprRS*::Cm$^r$ | This study |
| GG100 | MG1655 *msrP*::*lacZ* Δ*cusRS*::Cm$^r$ | This study |
| GG758 | MG1655 Δ*copA* Δ*cueO* | This study |
| GG769 | MG1655 Δ*copA* Δ*cueO* Δ*cusB*::Kan$^r$ | This study |
| GG770 | MG1655 Δ*copA* Δ*cueO* Δ*msrP*::Kan$^r$ | This study |
| LL1021 | MG1655 Δ*cueO* Δ*copA* Δ*cusF*::Kan$^r$ | This study |

**Table 2. Primers used in this study.** This table contains the information regarding the primers used in this study, including primer names and sequences.

| Name | Sequence (5' to 3') |
|---|---|
| cusC-EcoRI fwd- | CAGTGAATTCATGTCTCCTTGTAAACTTCTG |
| cusA-XhoI-rev | ACGCTCGAGTTATTTCCGTACCCGATGTCG |
| cusF-EcoRI fwd | CAGTGAATTCATGAAAAAAGCACTGCAAGTC |
| cusF-strep-HindIII rev | TTAAGCTTTTACTTTTCGAACTGCGGGTGGCTCCACTGGCTGACTTTAATATCCTGT |
| cusB-EcoRI fwd | CAGTGAATTCATGAAAAAAATCGCGCTTATTATCGGC |
| cusB-strep-HindIII rev | TTAAGCTTTTACTTTTCGAACTGCGGGTGGCTCCAATGCGCATGGGTAGCACTT |
| cusF fwd | ATCACCCCGCAGACGAAAATGAGTGAAATTAAAACCGGCGACAAAGTGG |
| cusF-M69-71I rev | TCATTTTCGTCTGCGGGGTGATGGTAAAGCGGATGGTGATCTCCGGCCAGT |
| cusF-M69-71Q rev | TCATTTTCGTCTGCGGGGTGATGGTAAAGCGCTGGGTCTGCTCCGGCCAGT |
| (62)5QTmsrP | TGATGACTTAACCCGTCGCT |
| (63)3QTmsrP | GCATCTGTTCCGGTGCATAA |
| (64)5QTmsrQ | TCGCCGCCTGTTAGGATTAT |
| (65)3QTmsrQ | AGTGAACGCTAAAGCAAGCA |
| (142)3QTstopyedX | TTAACTGCCACGATAGGTTGAATAC |
| (143)QTintyedX | ACGAATTAAGGCACTGTGGC |
| P1_Up_YedW(HprR) | TGTTTCTATAACATATGATTTATGGCATATTATTTTCATGGTGTAGGCTGGAGCTGCTTC |
| P2_Down_YedV(HprS) | TTTTCACGGTTAATTTATGGCGTACTGAAGCCCTATGTTACATATGAATATCCTCCTTAG |
| cusS_kan_rev | GGTTATAAAAGTTGCCGTTTGCTGAAGGATTAAGCGGGTAATGTGATAACCATATGAATATCCTCCTTA |
| cusR_kan_for | TCTGATCCCGCTACTCTAGAATTGCCCGGGCAACATGCGGAGGAAATATGGTGTAGGCTGGAGCTGCTTC |

*cusF-M69-71I rev* or *cusF-M69-71Q rev* (Table 2). The resulting PCR products were digested using *DpnI*, purified using the GeneJET PCR purification kit (Thermo Fisher) and transformed into *E. coli* DH5α. Three colonies were randomly selected from each transformation, and the plasmids were isolated using the GeneJET Plasmid Miniprep kit (Thermo Fisher). DNA sequencing was carried out to assess the fidelity of the mutagenesis reaction.

## RNA preparation, PCR from cDNA and qRT-PCR

Overnight cultures of wild-type cells (MG1655) were diluted to an $OD_{600nm}$ of 0.04 in fresh LB medium (5 ml) and grown aerobically at 37°C for 4 hours ($OD_{600nm} \approx 2$) in the presence or absence of $CuSO_4$ (500 μM). RNA was extracted with Maxwell 16 LEV miRNA Tissue Kit (Promega) according to the manufacturer's instructions and was subjected to an extra TURBO DNase (Invitrogen) digestion step to eliminate the contaminating DNA. The RNA quality was assessed by a tape station system (Agilent). RNA was quantified at 260 nm using a NanoDrop

**Table 3. Plasmids used in this study.** This table contains the information regarding the plasmids used in this study, including plasmid names, genotypes, description and source.

| Plasmid | Genotype and description | Source |
|---|---|---|
| pJF119-EH | $P_{lac}$ promoter, IPTG inducible, $Amp^R$ selection | [48] |
| pAV79 (pCusCFBA) | pJF119-EH-CusCFBA | This study |
| pAV54 (pCusF) | pJF119-EH-CusF(Strep-TagII) | This study |
| pAV83 | pJF119-EH-CusF(Strep-TagII) M47I/M49I | This study |
| pAV84 | pJF119-EH-CusF(Strep-TagII) M47Q/M49Q | This study |
| pAV67 (pCusB) | pJF119-EH-CusB(Strep-TagII) | This study |
| pECD735 | pASK-IBA3plus CusF-StrepTagII | [18] |
| pECD736 | pASK-IBA3plus CusF-StrepTagII M47I/M49I | [18] |
| pAV96 | pASK-IBA3plus CusF-StrepTagII M47Q/M49Q | This study |

1000 spectrophotometer (Thermo Fisher Scientific). Quantitative real-time PCR analyses were performed on a CFX96 Real-Time System (Bio-Rad) in a final volume of 15 µl with 0.5 µM final concentration of each primer using the following program: 98˚C for 2 min, then 45 cycles of 98˚C for 5 s, 56˚C for 10 s, and 72˚C for 1 s. A final melting curve from 65˚C to 95˚C was added to determine amplification specificity. The amplification kinetics of each product were checked at the end of each cycle by measuring the fluorescence derived from the incorporation of EvaGreen into the double-stranded PCR products using the SsoFast EvaGreen Supermix 2X Kit (Bio-Rad, France). The results were analyzed using Bio-Rad CFX Maestro software, version 1.1 (Bio-Rad, France). RNA were quantified and normalized to the 16S rRNA housekeeping gene. qRT-PCR for each condition were carried out in triplicate. All biological repeats were selected and reported. Amplification efficiencies for each primer pairs were between 75% and 100%. Primer pairs used for qRT-PCR are listed in Table 2.

## Immunoblot analysis of MsrP expression

To monitor MsrP expression levels after $CuSO_4$ treatment, overnight cultures of wild-type cells (MG1655) were diluted to an $OD_{600nm}$ of 0.04 in fresh LB medium (5 ml) and grown aerobically at 37˚C for 4 hours in the presence or absence of $CuSO_4$ (500 µM). Samples were suspended in Laemmli SDS sample buffer (2% SDS, 10% glycerol, 60 mM Tris-HCl, pH 7.4, 0.01% bromophenol blue), heated to 95˚C, and loaded onto an SDS-PAGE gel for immunoblot analysis. Protein amounts were standardized by taking into account the $OD_{600nm}$ values of the cultures. Western blotting was performed using standard procedures, with primary antibodies directed against MsrP (rabbit sera; Jean-François Collet laboratory), followed by a horseradish peroxidase (HRP)-conjugated anti-rabbit IgG secondary antibody (Promega). Chemiluminescence signals were detected using the GE ImageQuant LAS4000 camera (GE Healthcare Life Sciences).

## Copper and HOCl induction assays

The *msrP*::*lacZ*-containing strains (CH183 (WT), CH1000 (Δ*hprRS*) and GG100 (Δ*cusRS*)) were grown at 37˚C under agitation in M9 minimal medium supplemented with CASA (0.4% w/v). When cells reached an $OD_{600nm} \approx 0.2$, cultures were split into four plastic tubes, one control tube, one containing 150 µM HOCl, one containing 500 µM $CuSO_4$ and one supplemented with 150 µM HOCl and 500 µM $CuSO_4$, which were then incubated at an inclination of 90˚ with shaking at 37˚C. After 1 hour, 1 ml was harvested and the bacteria were resuspended in 1 ml of β-galactosidase buffer. Levels of β–galactosidase were measured as previously described [47].

## Copper survival assays

MG1655, BE107, GG758, GG769, GG770 and LL1021 cells were grown aerobically at 37˚C under agitation in 5 ml of M9 minimal medium (without CASA) in 50 ml conical polypropylene tubes (Sarstedt) with an inclination of 90˚. When cultures reached $OD_{600nm} \approx 0.1$, cells were harvested and diluted in phosphate buffered saline (PBS): 5 µL of 10-time serial dilutions were spotted onto M9 minimal medium-agar plates supplemented or not with $CuSO_4$ (12.5 and 20 µM in Fig 2A; 5 µM in Figs 2C and 3A-aerobic conditions and 3B; 1.5 µM in Fig 3A-anaerobic conditions; 25 µM in Figs 4 and 5B). Plates were incubated at 37˚C for 3 days. Ampicillin (50 µg/ml) and IPTG (50 or 100 µM) were added to solid and liquid media when required. For the anaerobic conditions, the plates were incubated at 37˚C for 4 days in a BD GasPak system.

## Protein expression and purification

Wild-type CusF and variants were expressed and purified as previously described by Pr. Dietrich H. Nies laboratory [18]. MG1655 cells harboring plasmids pECD735, pECD736, pAV96 and over-expressing wild-type CusF, CusF$^{M47I/M49I}$ and CusF$^{M47Q/M49Q}$ proteins respectively, were grown aerobically at 37˚C in LB supplemented with ampicillin (200 μg/ml). When cells reached an OD$_{600nm}$ of 0.8, expression was induced with anhydrotetracycline (200 μg/L final concentration) for 4 h at 30˚C. Periplasmic proteins were extracted and CusF was purified on a 5 ml Strep-Trap HP column (GE healthcare) equilibrated with buffer A (10 mM NaPi, pH 8.0, 500 mM NaCl). After washing the column with buffer A, CusF was eluted with buffer A supplemented with desthiobiotin (2.5 mM). The fractions containing CusF were checked using SDS PAGE and the clean fractions were pooled and desalted with buffer 40 mM MOPS, pH7, 150 mM NaCl.

## Protein oxidation and repair *in vitro*

Wild-type CusF protein (250 μM) was oxidized using H$_2$O$_2$ (50 mM) for 2 hours at 37˚C. The reaction was stopped by buffer exchange using Zeba Spin Desalting Columns, 7K MWCO, with 40 mM MOPS, pH7, 150 mM NaCl. The CusF$^{ox}$ protein formed was treated with MsrP enzyme in the presence of a reducing system to give the repaired form CusF$^{rep}$ by incubating 100 μM CusF$^{ox}$ for 2 hours at 30˚C in an anaerobic chamber with 4 μM purified MsrP, 10 mM benzyl-viologen and 10 mM dithionite. The reaction was stopped by buffer exchange using Zeba Spin Desalting Columns, 7K MWCO, with 40 mM MOPS, pH7, 150 mM NaCl.

## Mass spectrometry analysis

Samples were reduced and alkylated before digestion overnight with trypsin (to a final protease:protein ratio of 1:100) at 30˚C in 50 mM NH$_4$HCO$_3$ pH 8.0. Peptides were dissolved in solvent A (0.1% TFA in 2% ACN), immediately loaded onto a reverse-phase pre-column (Acclaim PepMap 100, Thermo Scientific) and eluted in backflush mode. Peptide separation was performed using a reverse-phase analytical column (Acclaim PepMap RSLC, 0.075 x 250 mm, Thermo Scientific) with a linear gradient of 4%-36% solvent B (0.1% FA in 98% ACN) for 36 min, 40%-99% solvent B for 10 min and holding at 99% for the last 5 min at a constant flow rate of 300 nl/min on an EASY-nLC 1000 UPLC system. Peptide analysis was carried out using an Orbitrap Fusion Lumos tribrid mass spectrometer (ThermoFisher Scientific). The peptides were subjected to NSI source followed by tandem mass spectrometry (MS/MS) in Fusion Lumos coupled online to the UPLC. Intact peptides were detected in the Orbitrap at a resolution of 120,000. Peptides were selected for MS/MS using an HCD setting of 30; ion fragments were detected in the Orbitrap at a resolution of 30,000. The electrospray voltage applied was 2.1 kV. MS1 spectra were obtained with an AGC target of 4E5 ions and a maximum injection time of 50 ms, and targeted MS2 spectra were acquired with an AGC target of 2E5 ions and a maximum injection time of 60 ms. For MS scans, the m/z scan range was 350 to 1800. The resulting MS/MS data were processed and quantified by LFQ (area under the curve) using Proteome Discoverer 2.4 against an *E. coli* K12 protein database obtained from Uniprot. Mass error was set to 10 ppm for precursor ions and 0.05 Da for fragment ions. Oxidation (+15.99 Da) on Met, pyro-Glu formation from Gln and Glu at the peptide terminus, N-terminal removal of Met and acetylation were considered as variable modifications.

## Fluorescence measurements

All experiments were performed at 25˚C using a SAFAS flx-Xenius 5117 spectrofluorimeter. Fluorescence measurements were carried out after dilution of wild-type CusF, CusF$^{ox}$, CusF$^{rep}$

or CusF$^{M47Q/M49Q}$ (1 μM final concentration) and equilibration for 5 min in 2 ml of a buffer containing 40 mM MOPS (pH = 7) and 150 mM NaCl. Increasing concentrations of AgNO$_3$ (0, 0.2, 0.4, 0.6, 0.8, 1, 1.5, 2, 3, 4, and 5 μM) were added and the emission fluorescence was scanned in the range of 300 to 384 nm, upon excitation at 284 nm. All spectra were corrected for buffer fluorescence with the same ligand concentration. Corrections for the inner-filter effect of the ligands were performed under the same conditions by using *N*-acetyltryptophana-mide (NATA). The CusF fluorescence spectrum is centred at 350 nm and NATA spectrum at 357 nm. Peak integration was carried out for each ligand concentration.

## Spectroscopic measurements

Far-UV circular dichroism spectroscopy of 13 μM CusF, CusF$^{M47Q/M49Q}$ and CusFox were recorded in 20 mM KH$_2$PO$_4$ (pH 7), 100 mM NaF buffer using a Jasco-815 spectropolarimeter at 25˚C. All spectra were buffer corrected.

## Statistical analysis

Mann-Whitney U tests were performed using the QI-Macros software (KnowWare Interna-tional, Inc., Denver, CO).

## Supporting information

**S1 Fig. Circular dichroism experiments.** Far-UV CD spectra of 13 μM CusF (black), CusF MQ (pink) and CusF ox (blue) were recorded at 25˚C.
(EPS)

## Acknowledgments

We thank the members of the Ezraty group for comments on the manuscript, advice and dis-cussions. Thanks to Pr. Dietrich H. Nies (Martin-Luther-Universitat Halle-Wittenberg) for providing CusF plasmids. We also thank M. Ilbert (BIP-CNRS), D. Byrne-Kodjabachian (IMM-CNRS) for the circular dichroism experiment, helpful suggestions, reagents and com-ments on the manuscript. Special thanks to the former Marseillaise Barras team (Team Barras 4 ever) and to Frederic Barras (now at the Institut Pasteur) for lab space, support and discussions.

## Author Contributions

**Conceptualization:** Alexandra Vergnes, Laurent Loiseau, Laurent Aussel, Benjamin Ezraty.

**Data curation:** Alexandra Vergnes, Camille Henry, Gaia Grassini, Laurent Loiseau, Yann Denis, Anne Galinier, Didier Vertommen, Laurent Aussel, Benjamin Ezraty.

**Formal analysis:** Alexandra Vergnes, Camille Henry, Gaia Grassini, Laurent Loiseau, Benja-min Ezraty.

**Funding acquisition:** Benjamin Ezraty.

**Investigation:** Alexandra Vergnes, Camille Henry, Gaia Grassini, Laurent Loiseau, Sara El Hajj, Yann Denis, Anne Galinier, Didier Vertommen, Laurent Aussel, Benjamin Ezraty.

**Methodology:** Alexandra Vergnes, Camille Henry, Gaia Grassini, Benjamin Ezraty.

**Project administration:** Benjamin Ezraty.

**Supervision:** Benjamin Ezraty.

**Writing – original draft:** Alexandra Vergnes, Benjamin Ezraty.

**Writing – review & editing:** Alexandra Vergnes, Benjamin Ezraty.

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
