## [Decision Letter · Decision Letter 0]

24 Apr 2022

Dear Dr Ezraty,

Thank you very much for submitting your Research Article entitled 'Periplasmic oxidized-protein repair during copper stress in E. coli: a focus on the metallochaperone CusF.' to PLOS Genetics.

The manuscript was fully evaluated at the editorial level and by independent peer reviewers. The reviewers appreciated the attention to an important topic but identified some concerns that we ask you address in a revised manuscript

We therefore ask you to modify the manuscript according to the review recommendations. Your revisions should address the specific points made by each reviewer.

[LINK]

Yours sincerely,

Sean Crosson

Associate Editor

PLOS Genetics

Lotte Søgaard-Andersen

Section Editor: Prokaryotic Genetics

PLOS Genetics

Reviewer's Responses to Questions

**Comments to the Authors:**

Reviewer #1: This manuscript by Vergnes et al describes the Cu-based regulation of the MsrP methionine sulfoxide reductase by the CusSR two component system and an investigation of how MsrP interacts with elements of the E. coli CusCFBA Cu efflux pump. The authors demonstrate that MsrP is able to repair oxidative damage to Cu-binding methionine residues in CusF, which restores a Cu-sensitivity phenotype observed in the absence of CusF or MsrP.

This is a well written and presented manuscript, and the experiments presented clearly support the story that is being told. There are only some minor issues, including that some parts of the manuscript are a bit vague, in particular the initial investigation of the regulation of msrPQ by CusSR lacks key information, including how many promoters are present in the huiH-msrPQ (there is a small gap of ~ 109 bp between huiH and msrPQ – is there another promoter?), where the proposed CusR binding site is located, relative to the genes, and there is no information about the fact that CusR and HprR target the same sequence until the discussion section. However, it would be interesting to explore competitive effects of Cu and HOCl on the activation of huiH and msrP. Additionally unclear is why huiH is expressed much higher levels than msrPQ.

The authors have just published a paper that contains some of this information (Hajj etal 2022), and it would be good to make use of it here.

Specific comments:

Author summary:

L38-40 – unclear, what is the ‘selective advantage’ (which property of methionine?), and why would it necessitate the methionines being highly susceptible to oxidative damage ( a similar passage is found in the discussion and this should be clarified in both instances.

Introduction:

L66 – please add some examples of which proteins involved in metal homeostasis were already identified as MsrP substrates.

L 75-77 – why is Cu more toxic under anaerobic conditions, and if that is so, why is that not seen in the current study? The findings from the anaerobic exposure experiments presented in the results should be discussed in a detailed context on increased anaerobic Cu toxicity mechanisms and why there is no effect observed here.

L103 – it would be good to at least mention what hype of enzyme huiH encodes early on, e.g. here.

If msrPQ is regulated by CusSR, why was it not detected in the studies cited (28,29) that identified huiH as being CusR regulated. (also comments for l.115)

Results:

L116 – please update – ‘…investigated the role of copper…’

L137 - .. msrP is part of the CusCR regulon.. – Should that be msrPQ? And is this statemen based on the findings from the previous paragraph or referencing another study?

L149 – it might be good to clearly state that the copA, cueO and cusB triple mutant strains was made as part of this work.

L172 – ‘..whereas the copper sensitivity phenotype etc etc.’ Sentence unclear. Why would the empty vector affect the copper phenotype of mutant strains (Fig.4)? Are there any reasons why this might be the case.

L190 – please update – should that be ‘ with the sulfur atom accessible’?

L195 ‘ mimetic’ – unclear, should that be mimic? Mimetic would be an adjective?

L211 please update– Met47 and Met49 are part of the same peptide

L218 – please update – we took advantage of..

L223- please clarify – was the fluorescence quenching different or did it only appear to be different, i.e. probably was not different? The results are quite clear with a different of 65%?

L230 “… that CusF oxidation disrupts…” – unclear. Your data suggests that the oxidation disrupts/ prevent Cu binding, however, I see no evidence that the mutations disrupt the protein, i.e. protein structure/ integrity?

L234 please update: ‘’.. the mutated M to Q forms of CusF in comparison..”

Discussion:

L249/250 unclear – labile site – labile in what sense? Please add details. Why would the labile nature of the methionine be a selective advantage, and why would this necessarily have to be ‘ balanced’ by high susceptibility to oxidation. Please clarify.

L262 – should that be ‘ reduces met-O in CusF…)

L263-265 – sentence unclear, please reword

L269 – unclear , what are the different Cu defence systems that are expressed depending on growth conditions, and how do they relate to the work reported here? Why would the existence of these systems lead to the conclusion that MsrP maintains CopA and CueO under other (presumably not aerobic?) growth conditions. Please add sufficient detail to make this clear.

L273 Please update – ‘copper participates in methionine…’

L288/289 – Where are the CusR/HprR boxes located relative to the huiH-msrPQ promoter? How many boxes and promoters are known for this gene region?

L300- …RclA prevents the formation…

Methods:

Throughout it is not fully clear when M9 or LB are used and why. Could that please be clarified?

L401 – Why can 500 microM Cu-sulfate be used in M9/CASA medium, i.e. the concentration used for LB, rather than M9?

L414 – what were the Cu concentrations used for this assay.

L442 – what concentration or protein to trypsin ratio was used here?

Figures:

Figure 4 – why does the presence of the empty vector change the Cu resistance phenotype of the strains in the absence of treatment? Is this experimental variation or was this observed in all replicates?

Why were different concentrations of IPTG used for Figure 4 and Figure 5C

Table 3 – please include a reference of pJF119-EH where details of the plasmid can be looked up or add information about the plasmid backbone and the inserts that were added.

Reviewer #2: This genetic and biochemical study thoroughly investigates the role of methionine oxidation in the metallo-chaperone CusF during copper stress. The authors found that the TCS CusRS activates the expression of the methionine sulfoxide reductase MsrPQ during copper stress, which appears to play a role for the functionality of the CusCFBA efflux pump under aerobic conditions. The authors further showed that CusF activity is impaired upon oxidation of two of its methionine residues, which can be rescued by MsrPQ. Overall, the study is well executed and includes all necessary controls. The manuscript is well-written and all conclusions stated by the authors are justified by the data presented. There is one major and a few minor comments the authors should address prior to publication:

Major comment:

For oxidation of CusF in vitro, the authors used 50 mM peroxide, which is a unphysiologically high concentration and likely leads to partial aggregation of CusF. This could potentially explain why the authors can only partially restore CusF activity in Fig. 5. If there is access to a circular dichroism spectrometer, the authors may want to examine the secondary structure of CusF(ox). It is likely that Copper stress generates ROS through the Fenton reaction, which could be quantified using ROS-specific fluorophores (i.e. H2DCFDA and others). This would give the authors an idea how much ROS is produced and that amount should be used to oxidize CusF.

Minor comments:

1) Lines 78-87: the authors introduce the different copper -defense systems in E. coli. Given that they provide a schematic overview anyways in Fig 2B, they could refer to it here already.

2) line 106: "... is induced during copper stress via CusSR. We then established a phenotypic..."

3) line 123: "....whether the TCS HprSR or CusSR regulate the expression ...."

4) The authors switch back and forth between different media (Fig. 1A: LB; Fig. 1C+D: M9/CASA; survival assays on plates: M9). What is their rationale for using different media?

5) Generally, the authors should revise their materials & methods section, which lacks information essential for successful reproduction of the data. Among others, examples are:

a)What is is the concentration of casamino acids added to M9 media?

b) qRT-PCRL What was the start OD when copper was added to E. coli? How many hours/minutes was the strain exposed to copper?

c) line 433: What was the CusF concentration that was oxidized?

6) Comparison of Fig. 2C/3A (5 uM copper) and Fig 3B (25 uM copper): 5-fold higher copper concentrations in Fig. 3B don't seem to substantially affect the survival of the ∆copA∆cueO strain. Do the authors have an explanation for the resistance?

7) The study shows that overexpression of CusB and to some degree the entire cus operon is toxic for the cell. Do the authors have an explanation why particularly CusB has such a strong effect?

Reviewer #3: The authors investigated the interplay between CusFCBA-mediated copper resistance and the periplasmic oxidized protein repair system MsrPQ with MsrP being a methionine sulfoxide reductase. The respective operon was expressed under control of the two-component regulatory system of the Cus system, CusSR, and HprRS, which senses reactive chlorine species. MsrP is required for copper resistance in the absence of the copper-exporting P-type APTase Cop A and the periplasmic Cu(I) oxidase CueO under oxic conditions, if reactive oxygen species were not removed by catalase, but not under anoxic conditions. Expression of cusCFBA mollified this effect, and the periplasmic copper-binding protein CusF was mainly responsible. In the central experiment of this publication, the authors demonstrated that surface-exposed Met residues, which were also required for copper binding, were targets of oxidative stress, mutated or oxidized CusF was not longer able to bind the Cu(I)-proxy Ag(I), and, most important, that MsrP was indeed able to repair oxidized CusF to restore metal-binding activity of CusF. The interplay between copper, oxidative stress, MsrPQ and Cus is an important observation with broad interest to the metals field.

1. Fig. 3. Copper is more toxic to E. coli under anoxic conditions, probably because it is predominantly present as Cu(I), than under oxic conditions. Fig. 3A, however, does not show any difference. Why? Secondly, “PBS” was no defined in the legend.

2. In the results, the headlines were not highlighted in any way, e.g. in blue as in the methods.

3. The scientific style is unusual. Nearly every sentence starts with “We”, “our” and reads more like a report of the last summer vacation for school and not like a scientific paper. It is customary that the authors step back behind their data. Moreover, simple past was not always used in the results. The data are nice, the description complete and concise, statistics done but I would strongly suggest to eliminate all “we”, “us”, “our” from the results part. It is OK for the last sentence of the introduction but not elsewhere.

**Have all data underlying the figures and results presented in the manuscript been provided?**

Reviewer #1: Yes

Reviewer #2: Yes

Reviewer #3: Yes

PLOS authors have the option to publish the peer review history of their article (what does this mean?). If published, this will include your full peer review and any attached files.

Reviewer #1: No

Reviewer #2: No

Reviewer #3: No

---

## [Decision Letter · Decision Letter 1]

9 Jun 2022

Dear Dr Ezraty,

We are pleased to inform you that your manuscript entitled "Periplasmic oxidized-protein repair during copper stress in E. coli: a focus on the metallochaperone CusF." has been editorially accepted for publication in PLOS Genetics. Congratulations!

Yours sincerely,

Sean Crosson

Associate Editor

PLOS Genetics

Lotte Søgaard-Andersen

Section Editor: Prokaryotic Genetics

PLOS Genetics

Comments from the reviewers (if applicable):

Reviewer's Responses to Questions

**Comments to the Authors:**

Reviewer #1: The authors have provided comprehensive responses to the points raised in the original review and have also updated the manuscript accordingly where necessary.

As stated in the original review this is a very interesting piece of work and the quality of the original manuscript was high. Following revision I have no further comments on this manuscript.

Reviewer #2: All comments have been addressed. Congratulations to the authors on a lovely story.

Reviewer #3: With exception of a different view concerning the many "we" (meaning: person comes before science), I am happy. A big step forward in understanding periplasmic copper homeostasis. All suggestions have been answered.

**Have all data underlying the figures and results presented in the manuscript been provided?**

Reviewer #1: Yes

Reviewer #2: Yes

Reviewer #3: Yes

PLOS authors have the option to publish the peer review history of their article (what does this mean?). If published, this will include your full peer review and any attached files.

Reviewer #1: No

Reviewer #2: **Yes: **Jan-Ulrik Dahl

Reviewer #3: No

**Data Deposition**

http://datadryad.org/submit?journalID=pgenetics&manu=PGENETICS-D-22-00411R1

**Press Queries**

---

## [Editor Report · Acceptance letter]

29 Jun 2022

PGENETICS-D-22-00411R1 

Periplasmic oxidized-protein repair during copper stress in E. coli: a focus on the metallochaperone CusF. 

Dear Dr Ezraty, 

We are pleased to inform you that your manuscript entitled "Periplasmic oxidized-protein repair during copper stress in E. coli: a focus on the metallochaperone CusF." has been formally accepted for publication in PLOS Genetics! Your manuscript is now with our production department and you will be notified of the publication date in due course.

With kind regards,

Anita Estes

PLOS Genetics

On behalf of:
